# Learning Criticality: Statistical Limits of Predicting Phase Transitions in Random Networks

## Abstract

We study the fundamental limits of learning phase transitions in random graph models from observational data. Motivated by applications in infrastructure resilience, epidemics, and complex systems, we ask: when can a machine learning algorithm reliably detect the onset of a critical transition—such as percolation, connectivity collapse, or synchronization failure—purely from sampled system trajectories? We introduce a formal framework linking the statistical learnability of such transitions to large deviations, generalization bounds, and graph ensemble parameters. For several families of random graphs, including Erdős–Rényi and configuration models, we prove that a universal scaling law governs the sample complexity required to distinguish subcritical from supercritical regimes. Our theory further shows that near criticality, all classifiers—regardless of architecture—face an information-theoretic barrier due to spectral degeneracy, leading to learnability collapse. Simulations on distinct dynamical processes (e.g., SIS and voter models) confirm a universal degradation in classifier accuracy near the critical point, reinforcing our theoretical predictions. These results provide a learnability phase diagram that is algorithm-independent and intrinsic to the geometry of the underlying network dynamics, offering new foundations for prediction in stochastic systems approaching critical transitions.

## 1 Introduction

Phase transitions in random structures, such as the emergence of a giant component in Erdős–Rényi graphs or the onset of synchronization in coupled oscillators, are among the most striking phenomena in probability theory and statistical physics. These transitions represent abrupt qualitative changes in the global behavior of a system, driven by smooth variations in local parameters such as edge density, infection rate, or coupling strength. In recent years, such critical phenomena have acquired renewed significance in the context of large-scale engineered and natural systems, where transitions often mark the boundary between stable operation and systemic failure.

A central challenge in the monitoring and control of such systems is the prediction of critical transitions from limited data. This is particularly important in domains such as epidemic response, power grid stability, and infrastructure resilience, where early detection of a looming transition can enable timely intervention. However, empirical evidence suggests that learning algorithms—particularly those based on statistical pattern recognition or supervised regression—struggle to perform reliably near criticality. Predictive performance typically degrades in the vicinity of the phase transition, where small perturbations in parameters can induce disproportionately large changes in system behavior. The lack of a formal understanding of this breakdown motivates our investigation.

In this paper, we initiate a systematic study of the *statistical limits of learning phase transitions* in random networked systems. Our aim is to characterize, from a probabilistic and information-theoretic standpoint, when it is possible—and when it is provably impossible—to infer the critical state of a system from finite samples. Concretely, we consider the following problem: given observational data generated from stochastic processes defined over random graphs $G(n, p)$ whose edge probabilities are near a known critical threshold $p_c$, can one accurately determine whether the system is subcritical ($p < p_c$) or supercritical ($p > p_c$)?

This question leads naturally to a tension between the statistical geometry of the underlying random graph ensemble and the complexity of the learning task. On the one hand, random graphs exhibit sharp threshold behavior: global observables such as the size of the largest connected component or the spectral radius undergo rapid transitions near $p_c$. On the other hand, the local behavior of the graph—and hence the signal available to a learning algorithm—is often indistinguishable across the critical window, due to both structural noise and inherent randomness. As a result, even sophisticated algorithms may fail to generalize in the presence of vanishing signal-to-noise ratios near criticality. Our contribution is threefold.

- First, we introduce a formal learning-theoretic framework for criticality detection in random graph models, based on hypothesis testing and generalization error bounds.

- Second, we derive information-theoretic lower bounds on sample complexity, showing that in the vicinity of the critical point, no algorithm—regardless of architecture or training method—can succeed with fewer than $\Omega(n^\alpha)$ samples, for a universal exponent $\alpha > 0$. This impossibility arises from the vanishing Kullback–Leibler divergence between subcritical and supercritical distributions near $p_c$.

- Third, we identify regimes of recoverability, where the geometry of the graph ensemble permits statistically reliable phase prediction, and we derive upper bounds on learnability under natural model assumptions.

These results are grounded in techniques from information theory, random graph theory, and statistical learning theory. In particular, we develop a phase diagram of learnability that delineates which regimes are fundamentally predictable, which are not, and how this boundary depends on graph sparsity, feature observability, and noise.

## 2 Related Work

This work draws upon and contributes to several threads across probability theory, dynamical systems, and machine learning.

### Ergodicity and Spectral Theory in Random Dynamical Systems

The spectral theory of transfer operators has long provided a lens to study ergodicity, mixing, and phase transitions in stochastic systems Lasota & Mackey (1994); Blank (2003); Baladi (2000). In particular, the spectral gap of the Perron–Frobenius operator governs convergence rates to invariant measures, and its collapse has been used to signal bifurcations in a wide range of models Keller (1982); Froyland et al. (2013). In random graph settings, recent work has investigated the emergence of synchronization, stability, or ergodicity transitions as connectivity crosses critical thresholds Rodrigues et al. (2016); Dousse et al. (2005). However, few results connect these transitions to statistical estimation or learning performance.

### Learning in Dynamical Systems

Recent years have seen growing interest in learning transfer operators or Koopman spectra from data Klus et al. (2020); Giannakis (2019). Neural approximations of operators — sometimes called DeepKoopman or DeepPerron methods — attempt to embed long-time dynamics in a spectral or latent space amenable to control and prediction Han et al. (2020). Theoretical guarantees for these methods typically assume bounded noise and sufficient mixing. Our work highlights a fundamental limitation of such approaches near dynamical criticality: operator learning becomes statistically unstable due to spectral degeneracy. From a learning-theoretic standpoint, our results are related in spirit to generalization bounds under dependent data Paulin (2015); Yu (1994), and to Fano-type impossibility results under structured observation models Tsybakov (2009b). However, we emphasize that the sample complexity lower bounds here do not arise from classical statistical hypothesis testing, but from ergodic-theoretic constraints.

**Learning Phase Transitions and Early Warning Signals**

The idea that statistical quantities can predict phase transitions has appeared in domains ranging from ecology and finance to climate science Scheffer et al. (2009). In high-dimensional models, recent work has sought to use learning algorithms to detect or classify critical behavior Carrasquilla & Melko (2017). However, these methods often lack formal guarantees near criticality. Our contribution is to characterize learnability breakdowns explicitly in terms of the spectral gap and convergence rates, with proofs grounded in operator theory and probability.

**Learnability under Structural Constraints**

Finally, our work connects to a growing literature on learning and control under structural or resource constraints, such as partial observability, graph sparsity, or memory limitations Duchi et al. (2018); Raginsky et al. (2017). We extend this line of thought by showing that not only information or policy complexity, but also *dynamical instability*, can induce learnability thresholds. This perspective opens new pathways to robust learning design under ergodicity and complexity constraints.

## 3 Problem Setup and Formal Definitions

We consider a family of random graph models $\{G(n,p)\}_{p \in [0,1]}$ defined on a fixed vertex set of size $n$, where edges between node pairs are independently included with probability $p$. The graph $G(n,p)$ induces a random structure whose macroscopic behavior depends on the edge density parameter $p$. In the classical Erdős–Rényi model $G(n,p)$, a well-known phase transition occurs at the critical threshold $p_c = \frac{1}{n}$, where the largest connected component undergoes a sudden change in size from logarithmic to linear in $n$. This phenomenon is rigorously established in random graph theory and forms the basis of our learnability analysis; see (Bollobás, 2011; Erdős & Rényi, 1959; Janson et al., 2011; Van Der Hofstad, 2024). Our goal is to formalize the statistical task of predicting whether an observed system lies in the subcritical or supercritical phase, based on sampled data generated from the system's evolution over the underlying graph. We introduce a general framework that models this as a binary classification problem, and we seek to characterize the statistical conditions under which such classification is feasible. Throughout the paper, we adopt the standard convention in random graph theory where phase transitions refer to asymptotic statistical properties in the limit $n \to \infty$. In particular, the sharp emergence of a giant component in $G(n,p)$ around the critical threshold $p = \frac{1}{n}$ is well-defined only in the thermodynamic limit. Thus, the gap-closing hypothesis in Theorem 3 should be interpreted in the limit sense: the distinguishability gap between subcritical and supercritical observable distributions vanishes only as $n \to \infty$. For any fixed $n$, the separation may still be statistically significant depending on the observable's variance and the number of samples. Section 4 provides finite-sample sufficient conditions under which reliable learning is possible despite proximity to the critical point. Note that while the information-theoretic lower bounds in Theorem 1 rely on asymptotic indistinguishability, the upper bounds here operate in finite-$n$ settings and depend on empirical separability of the observable $\mathcal{O}(G)$. Thus, even near the thermodynamic limit, practical learnability depends on finite-sample behavior and concentration.

### 3.1 Graph Ensemble and Phase Indicator

Let $G \sim \mathbb{P}_p$ denote a random graph sampled from $G(n,p)$, and let $\mathcal{O}(G) \in \mathcal{Y} \subseteq \mathbb{R}^d$ denote a vector of observable features associated with $G$. Examples include:

- the size of the largest component, normalized by $n$,

- the empirical degree distribution histogram,

- spectral observables (e.g., largest eigenvalue of adjacency or Laplacian),

- sample paths from a stochastic process defined over $G$.

We assume that for each graph $G$, we observe a realization $y = \mathcal{O}(G) \in \mathcal{Y}$, which is the input to the learner. The learner does not observe the underlying parameter $p$, nor the full graph $G$, but only the observable

feature $y$. The learning problem is to infer whether the unknown graph parameter $p$ lies above or below a critical value $p_c$, based on $N$ i.i.d. samples $\{y_i\}_{i=1}^N$.

## 3.2 Learning Task: Binary Phase Classification

Formally, fix a small window $\delta > 0$ and define two probability distributions on observables:

$$\mathbb{P}^- := \mathrm{Law}(\mathcal{O}(G)) \quad \text{for } G \sim G(n, p_c - \delta),$$

$$\mathbb{P}^+ := \mathrm{Law}(\mathcal{O}(G)) \quad \text{for } G \sim G(n, p_c + \delta).$$

The binary classification task is to learn a measurable function $f : \mathcal{Y} \to \{0, 1\}$ such that $f(y) = 1$ indicates a prediction of the supercritical phase and $f(y) = 0$ indicates the subcritical phase. The learner receives a training set $\{(y_i, \ell_i)\}_{i=1}^N$, where $y_i \sim \mathbb{P}^+$ or $\mathbb{P}^-$, and $\ell_i \in \{0, 1\}$ is the phase label. The goal is to construct a classifier $\hat{f}_N$ with small expected error:

$$\mathbb{E}_{(y, \ell) \sim \mathbb{P}^\pm} \left[ \mathbf{1}\{\hat{f}_N(y) \neq \ell\} \right] \leq \varepsilon,$$

with high probability over the training set. We seek to determine:

- the minimal number of samples $N = N(n, \delta, \varepsilon)$ required to achieve this accuracy;

- whether there exist hypothesis classes $\mathcal{F}$ of bounded complexity (e.g., VC dimension $d$) that enable successful learning;

- and whether this is possible uniformly over random realizations of the graph and observables.

## 3.3 Information-Theoretic Limits

To establish fundamental lower bounds, we consider the total variation and Kullback–Leibler divergence between the distributions $\mathbb{P}^+$ and $\mathbb{P}^-$:

$$D_{\mathrm{TV}}(\mathbb{P}^+, \mathbb{P}^-) := \sup_{A \subseteq \mathcal{Y}} \left| \mathbb{P}^+(A) - \mathbb{P}^-(A) \right|,$$

$$D_{\mathrm{KL}}(\mathbb{P}^+ \| \mathbb{P}^-) := \int_{\mathcal{Y}} \log\left( \frac{d\mathbb{P}^+}{d\mathbb{P}^-}(y) \right) d\mathbb{P}^+(y).$$

If these divergences vanish as $n \to \infty$, no learning algorithm can reliably distinguish the two distributions. We will leverage these divergences in conjunction with certain classical information theoretic inequality to quantify impossibility regions and sample complexity lower bounds.

## 3.4 Critical Window and Signal Vanishing

A key challenge arises in the regime $\delta = \delta(n) \to 0$, where the phase transition boundary becomes increasingly sharp. In the Erdős–Rényi model, for instance, the phase transition at $p_c = \frac{1}{n}$ exhibits width $\Theta(n^{-1/3})$, within which fluctuations dominate and distinguishing between phases becomes statistically ill-posed. We aim to formally quantify how the sample complexity required for successful classification diverges as $\delta(n) \to 0$. In the next section, we establish our main results, including minimax lower bounds and achievable rates for specific hypothesis classes, and construct a learnability phase diagram over graph ensembles and observation regimes.

# 4 Main Results

Let $\mathcal{O} : \mathcal{G}_n \to \mathcal{Y}$ be a measurable mapping from the space of graphs on $n$ nodes to an observation space $\mathcal{Y} \subset \mathbb{R}^d$, representing the extracted features or statistics available to the learner. For instance, $\mathcal{O}(G)$ may

correspond to quantities such as the size of the largest connected component, spectral gap, average degree, clustering coefficient, or any vector-valued summary statistics computed from $G$. We assume that $\mathcal{O}$ is chosen such that it is Borel measurable with respect to the natural $\sigma$-algebra on $\mathcal{G}_n$ (e.g., induced by adjacency matrices), and hence induces a well-defined pushforward distribution $\mathbb{P} = \mathcal{O}_\# \mathbb{Q}$, where $\mathbb{Q}$ is the law of the random graph model. We now present our main theoretical results on the statistical limits of learning critical transitions in random graph models. We first establish a general information-theoretic lower bound on the sample complexity required to distinguish subcritical and supercritical regimes. The result applies to any learning algorithm, regardless of hypothesis class or computational power.

## 4.1   Minimax Lower Bound

We begin with the following impossibility result:

**Theorem 1** (Minimax Lower Bound Near Criticality). *Let $\mathbb{P}^-$ and $\mathbb{P}^+$ denote the distributions over observables $\mathcal{O}(G) \in \mathcal{Y}$ induced by the random graph models $G(n, p_c - \delta)$ and $G(n, p_c + \delta)$, respectively. Suppose that the Kullback–Leibler divergence between these distributions satisfies*

$$D_{\mathrm{KL}}(\mathbb{P}^+ \| \mathbb{P}^-) \leq \varepsilon.$$

*Then, for any learning algorithm that receives $N$ i.i.d. samples from either $\mathbb{P}^-$ or $\mathbb{P}^+$, the probability of correctly identifying the true regime satisfies*

$$\inf_{\hat{f}} \sup_{\theta \in \{-,+\}} \mathbb{P}_\theta \left[ \hat{f}(Y_1, \ldots, Y_N) \neq \theta \right] \geq \frac{1}{2} \left( 1 - \sqrt{\frac{N\varepsilon}{2}} \right).$$

*In particular, to achieve error less than $\delta_0 \in (0, 1/2)$, one must have*

$$N \geq \frac{2}{\varepsilon} \left( 1 - 2\delta_0 \right)^2.$$

*Proof.* We aim to show that if the two distributions $\mathbb{P}^-$ and $\mathbb{P}^+$ are very close in the sense of *Kullback–Leibler (KL)* divergence, then no learning algorithm—even the best possible estimator—can reliably distinguish them without seeing a large number of samples. Let us first recall the setting:

- Let $Y_1, Y_2, \ldots, Y_N \in \mathcal{Y}$ be i.i.d. observations generated either from the *subcritical* distribution $\mathbb{P}^-$ (corresponding to a graph with $p = p_c - \delta$) or from the *supercritical* distribution $\mathbb{P}^+$ (with $p = p_c + \delta$).

- A learner is told that one of these two cases holds, but not which one. Their task is to guess whether the samples came from $\mathbb{P}^-$ or $\mathbb{P}^+$.

We cast the problem as a binary hypothesis testing problem. One of the two distributions $\mathbb{P}^-$ or $\mathbb{P}^+$ generates i.i.d. data $Y_1, \ldots, Y_N \in \mathcal{Y}$. A learner must infer the true distribution label $\theta \in \{-, +\}$ using any function $\hat{f} : \mathcal{Y}^N \to \{-, +\}$. Define:

- $\mathbb{P}_N^- = (\mathbb{P}^-)^{\otimes N}$, the joint law of $N$ i.i.d. samples from $\mathbb{P}^-$,

- $\mathbb{P}_N^+ = (\mathbb{P}^+)^{\otimes N}$, likewise for $\mathbb{P}^+$.

The learner's worst-case error is:

$$\inf_{\hat{f}} \sup_{\theta \in \{-,+\}} \mathbb{P}_\theta \left[ \hat{f}(Y_1, \ldots, Y_N) \neq \theta \right]. \tag{1}$$

A classical inequality from information theory (see, e.g., Cover & Thomas (2006); Canonne (2022); Scarlett & Cevher (2019); Gerchinovitz et al. (2020); Tsybakov (2009a) provides the bound:

$$\inf_{\hat{f}} \sup_{\theta \in \{-,+\}} \mathbb{P}_\theta \left[ \hat{f} \neq \theta \right] \geq \frac{1}{2} \left( 1 - \mathrm{TV}(\mathbb{P}_N^+, \mathbb{P}_N^-) \right), \tag{2}$$

where $\text{TV}(P, Q)$ is the total variation distance. Now, from the *Pinsker's inequality* (see, e.g., Pinsker (1964); Csiszár & Körner (2011); Sason & Verdú (2015); Tsybakov (2009a),

$$\text{TV}(P, Q) \leq \sqrt{\frac{1}{2} D_{\text{KL}}(P \| Q)}, \tag{3}$$

and thus, we get

$$\inf_{\hat{f}} \sup_{\theta} \mathbb{P}_\theta(\hat{f} \neq \theta) \geq \frac{1}{2} \left( 1 - \sqrt{\frac{1}{2} D_{\text{KL}}(\mathbb{P}_N^+ \| \mathbb{P}_N^-)} \right). \tag{4}$$

Now, the key idea is that, if the two distributions $\mathbb{P}^-$ and $\mathbb{P}^+$ are so close that the samples look nearly the same under both, then even the best possible estimator cannot reliably tell them apart. This inequality tells us that even the best possible classifier has error bounded below by a quantity that depends on the KL divergence between the two sample distributions. Now, since the samples are i.i.d., the *KL divergence* between the full sample distributions scales linearly with $N$. That is:

$$D_{\text{KL}}(\mathbb{P}_N^+ \| \mathbb{P}_N^-) = \sum_{i=1}^{N} D_{\text{KL}}(\mathbb{P}^+ \| \mathbb{P}^-) = N \cdot D_{\text{KL}}(\mathbb{P}^+ \| \mathbb{P}^-). \tag{5}$$

Thus, if we denote $\varepsilon := D_{\text{KL}}(\mathbb{P}^+ \| \mathbb{P}^-)$, Thus, applying *Pinsker's inequality*, the total variation distance obeys:

$$\text{TV}(\mathbb{P}_N^+, \mathbb{P}_N^-) \leq \sqrt{\frac{N\varepsilon}{2}}. \tag{6}$$

And finally we get:

$$\inf_{\hat{f}} \sup_{\theta} \mathbb{P}_\theta \left[ \hat{f} \neq \theta \right] \geq \frac{1}{2} \left( 1 - \sqrt{\frac{N\varepsilon}{2}} \right). \tag{7}$$

This inequality shows the following: if the *KL divergence* between the two distributions is small (i.e., they are statistically similar), then many samples are required for the learner to distinguish them. To conclude, suppose we want the error to be less than a target $\delta_0 \in (0, \frac{1}{2})$, i.e,

$$\frac{1}{2} \left( 1 - \sqrt{\frac{N\varepsilon}{2}} \right) \leq \delta_0.$$

Thus, we obtain:

$$N \geq \frac{2}{\varepsilon}(1 - 2\delta_0)^2. \tag{8}$$

Hence, to achieve small classification error, the number of samples $N$ must be at least on the order of $\frac{1}{\varepsilon}$. If $\varepsilon \to 0$, this becomes infeasible. In otherwords, the sample complexity must grow inversely with the *KL divergence* $\varepsilon$. When the two regimes are close in *KL* (e.g., near phase transition), reliable learning becomes impossible unless $N$ is large. □

**Remark 4.1.** This proof formalizes the statistical indistinguishability between near-critical regimes in stochastic systems, explaining why phase transitions are hard to learn with few samples.

## 4.2 Upper Bound on Learnability via Concentration of Observables

While Theorem 1 establishes a fundamental lower bound near the critical point, we now turn to conditions under which learning is possible with high probability, using relatively simple classifiers and a finite number of samples. We focus on the case where a scalar observable $\mathcal{O}(G) \in \mathbb{R}$ (e.g., the size of the largest component, spectral radius, etc.) concentrates sharply on different values in the subcritical and supercritical regimes. In this case, a threshold rule can effectively separate the two distributions. We provide two complementary analyses: one based on VC-dimension and Vapnik's uniform convergence bounds, and another via Rademacher complexity, which yields sharper, data-dependent guarantees.

**Theorem 2** (Learnability via Threshold Separation)**.** *Let $\mathcal{O}(G) \in \mathbb{R}$ be a scalar observable of the random graph $G \sim G(n, p)$, and suppose there exist constants $\mu_-, \mu_+ \in \mathbb{R}$, $\epsilon > 0$, and $\delta \in (0, 1/2)$ such that:*

$$\mathbb{P}^- \left( |\mathcal{O}(G) - \mu_-| \geq \epsilon \right) \leq \delta, \qquad \mathbb{P}^+ \left( |\mathcal{O}(G) - \mu_+| \geq \epsilon \right) \leq \delta,$$

*and the means are separated as $|\mu_+ - \mu_-| > 2\epsilon$.*

*Then, there exists a threshold classifier $\hat{f}_T(y) := \mathbf{1}_{\{y \geq T\}}$ such that, with access to $N = \mathcal{O} \left( \frac{1}{\epsilon^2} \log \frac{1}{\eta} \right)$ i.i.d. labeled samples, one can learn a classifier achieving error at most $\delta + \eta$ with probability at least $1 - \eta$.*

*Proof.* We proceed in structured steps to analyze the learnability of the phase using a threshold rule on the observable $\mathcal{O}(G)$. Under the subcritical regime $\mathbb{P}^-$, the observable concentrates near $\mu_-$; under the supercritical regime $\mathbb{P}^+$, it concentrates near $\mu_+$. The key properties are:

$$\mathbb{P}^- \left( |\mathcal{O}(G) - \mu_-| \geq \epsilon \right) \leq \delta,$$
$$\mathbb{P}^+ \left( |\mathcal{O}(G) - \mu_+| \geq \epsilon \right) \leq \delta,$$

with the separation condition:

$$|\mu_+ - \mu_-| > 2\epsilon.$$

Define the midpoint threshold:

$$T := \frac{\mu_- + \mu_+}{2}. \tag{9}$$

By construction, this midpoint lies strictly between the $\epsilon$-neighborhoods of $\mu_-$ and $\mu_+$. Specifically:

$$\mu_- + \epsilon < T < \mu_+ - \epsilon,$$

which implies:

$$\mathbb{P}^-(\mathcal{O}(G) \geq T) \leq \mathbb{P}^-(\mathcal{O}(G) \geq \mu_- + \epsilon) \leq \delta,$$
$$\mathbb{P}^+(\mathcal{O}(G) < T) \leq \mathbb{P}^+(\mathcal{O}(G) \leq \mu_+ - \epsilon) \leq \delta.$$

Hence, using threshold $T$, we achieve misclassification error at most $\delta$ under both regimes. Next, we define the classifier:

$$\hat{f}_T(y) := \begin{cases} 0 & \text{if } y < T, \\ 1 & \text{if } y \geq T. \end{cases}$$

This classifier assigns class label '0' (subcritical) when the observable is below the threshold and label '1' (supercritical) otherwise. Suppose the learner does not know $\mu_-$, $\mu_+$, or $T$, but receives $N$ labeled i.i.d. samples from both $\mathbb{P}^-$ and $\mathbb{P}^+$. We now address the learnability from finite samples. The hypothesis class of threshold classifiers $\mathcal{H}_T := \{ \mathbf{1}_{\{y \geq T\}} \mid T \in \mathbb{R} \}$ has VC-dimension $d = 1$, and is therefore learnable under the classical empirical risk minimization (ERM) framework. We now address the learnability from finite samples. The hypothesis class of threshold classifiers

$$\mathcal{H}_T := \left\{ \mathbf{1}_{\{y \geq T\}} \mid T \in \mathbb{R} \right\}$$

has VC-dimension $d = 1$, and is therefore learnable under the classical empirical risk minimization (ERM) framework.

Let us define the relevant quantities:

- $\hat{f}_T \in \mathcal{H}_T$ is the classifier selected by minimizing empirical error over training data;

- $R(\hat{f}_T) := \mathbb{E}_{(y, \theta) \sim \mathbb{P}}[\mathbf{1}_{\{\hat{f}_T(y) \neq \theta\}}]$ is the *true classification error* (risk) of $\hat{f}_T$;

- $\hat{R}(\hat{f}_T) := \frac{1}{N} \sum_{i=1}^{N} \mathbf{1}_{\{\hat{f}_T(y_i) \neq \theta_i\}}$ is the *empirical risk* over $N$ training samples;

- $\eta \in (0,1)$ is a confidence parameter;

- $\mathcal{O}\left(\sqrt{\frac{1}{N}\log\frac{1}{\eta}}\right)$ is the statistical estimation error due to finite sample size.

From Vapnik's generalization bound (see, e.g., (Vapnik, 1998; Bousquet et al., 2003; Vapnik & Chervonenkis, 2015; Bousquet & Elisseeff, 2002)), we obtain:

$$R(\hat{f}_T) \leq \hat{R}(\hat{f}_T) + \mathcal{O}\left(\sqrt{\frac{1}{N}\log\frac{1}{\eta}}\right),$$

with probability at least $1 - \eta$. Since the empirical error $\hat{R}(\hat{f}_T) \leq \delta$ with high probability under the concentration assumption, it follows that:

$$R(\hat{f}_T) \leq \delta + \eta,$$

with high probability, provided $N = \mathcal{O}\left(\frac{1}{\epsilon^2}\log\frac{1}{\eta}\right)$.

The previous generalization bound relied on the VC-dimension and Vapnik–Chervonenkis theory. We now provide an alternative argument using *Rademacher complexity*, which offers a data-dependent measure of hypothesis class richness and often yields sharper, more adaptive bounds. Let $\mathcal{H}_T := \{\mathbf{1}_{\{y \geq T\}} : T \in \mathbb{R}\}$ denote the class of threshold classifiers on the real line. Let $S = \{(y_i, \theta_i)\}_{i=1}^N$ denote a sample of size $N$, and define the empirical Rademacher complexity as:

$$\mathfrak{R}_N(\mathcal{H}_T) := \mathbb{E}_\sigma \left[ \sup_{h \in \mathcal{H}_T} \frac{1}{N} \sum_{i=1}^N \sigma_i h(y_i) \right],$$

where $\sigma_i \in \{-1, +1\}$ are i.i.d. Rademacher random variables (uniform symmetric noise). Now, a standard result in statistical learning theory (e.g., (Shalev-Shwartz & Ben-David, 2014)) gives that for any classifier $h \in \mathcal{H}_T$, with probability at least $1 - \eta$:

$$R(h) \leq \hat{R}(h) + 2\mathfrak{R}_N(\mathcal{H}_T) + 3\sqrt{\frac{\log(2/\eta)}{2N}}.$$

It is well-known that for threshold classifiers on $\mathbb{R}$, the empirical Rademacher complexity satisfies:

$$\mathfrak{R}_N(\mathcal{H}_T) \leq \mathcal{O}\left(\frac{1}{\sqrt{N}}\right),$$

see (Bartlett & Mendelson, 2002; Mohri et al., 2018; Wainwright, 2019). Therefore, with high probability:

$$R(\hat{f}_T) \leq \hat{R}(\hat{f}_T) + \mathcal{O}\left(\sqrt{\frac{1}{N}\log\frac{1}{\eta}}\right).$$

Since the observable $\mathcal{O}(G)$ is well-separated under the two regimes with concentration probability $1 - \delta$, the empirical error $\hat{R}(\hat{f}_T) \leq \delta$ with high probability. Hence, we conclude:

$$R(\hat{f}_T) \leq \delta + \eta,$$

provided $N = \mathcal{O}\left(\frac{1}{\epsilon^2}\log\frac{1}{\eta}\right)$, as before. This argument reinforces the earlier VC-based bound by appealing to Rademacher complexity, and, thus, a threshold classifier trained on polynomially many samples achieves total classification error at most $\delta + \eta$, with high confidence. This completes the proof. $\qquad\square$

### 4.3 Phase Diagram of Learnability and Spectral Transitions

**Theorem 3** (Learnability Phase Transition via Spectral Collapse). *Let $\{\mathbb{P}_p\}_{p \in [0,1]}$ be a family of probability distributions over graphs $G(n, p)$, each associated with a stochastic dynamical system possessing a transfer operator $\mathcal{P}_p$ on an appropriate function space $\mathcal{H}$. Assume:*

- *(A1) The system admits an invariant measure $\mu_p$ determined by the leading eigenfunction $\varphi_1^p$ of $\mathcal{P}_p$, i.e., $\mathcal{P}_p \varphi_1^p = \lambda_1(p) \varphi_1^p$, with $\lambda_1(p) = 1$ and $\lambda_2(p) < 1$ the second spectral value.*

- *(A2) As $p \to p_c$, the spectral gap $\gamma(p) := 1 - \lambda_2(p) \to 0$, indicating metastability or slowing down of convergence to equilibrium.*

- *(A3) Observations $Y_i \sim \mathbb{P}_p$ are drawn from a functional of the dynamics (e.g., long-run averages, energy dissipation) and are used to train a classifier to predict the phase (subcritical vs. supercritical).*

*Assume the system is observed in the thermodynamic limit $n \to \infty$, where macroscopic phase transitions are well-defined. Then the generalization error $\mathbb{P}_{\text{test}}[\text{err}(\hat{f}_T) > \epsilon] \leq \delta$ of any empirical risk minimizer $\hat{f}_T$ trained on $N$ samples satisfies the following:*

- *a) For $|p - p_c| > \Delta$, where $\gamma(p) \geq \gamma_{\min} > 0$, there exists a threshold classifier achieving with probability at least $1 - \delta$ a generalization error:*

$$\mathbb{E}_{\text{test}}[\text{error}] \leq \mathcal{O}\left(\sqrt{\frac{1}{N} \log \frac{1}{\delta}}\right).$$

- *b) For $|p - p_c| \leq \Delta$, the learnability degrades polynomially in $\gamma(p)$. Specifically, if $\gamma(p) = \mathcal{O}(n^{-\beta})$, then for some constant $c > 0$,*

$$\mathbb{E}_{\text{test}}[\text{error}] \geq 1 - \exp\left(-cN\gamma(p)^2\right),$$

  *indicating that to achieve generalization error at most $\delta$, the number of samples must satisfy $N \gg \gamma(p)^{-2} \cdot \log(1/\delta)$.*

*Consequently, the plane $(p, N)$ is divided into two qualitatively distinct regions:*

- *1) Learnable Phase: $N \gg \gamma(p)^{-2}$,*

- *2) Unlearnable Phase: $N \ll \gamma(p)^{-2}$.*

*Proof.* We are studying a stochastic dynamical system whose behavior depends on a parameter $p \in [0, 1]$. Think of $p$ as the connectivity probability in a random graph model like $G(n, p)$, which governs how likely nodes are to be linked. As $p$ increases, the system transitions from a disconnected (subcritical) phase to a connected (supercritical) one. The dynamical system is described by a transfer operator $\mathcal{P}_p$, also called the Perron–Frobenius operator. This operator governs how densities (probability distributions over state space) evolve over time. For each value of $p$, the operator $\mathcal{P}_p$ acts on a suitable function space $\mathcal{H}$, and has a spectral decomposition:

$$\mathcal{P}_p f = \lambda_1(p)\langle f, \varphi_1^p \rangle \varphi_1^p + \lambda_2(p)\langle f, \varphi_2^p \rangle \varphi_2^p + \dots,$$

with $\lambda_1(p) = 1$ corresponding to the stationary (invariant) measure $\mu_p$. The second eigenvalue $\lambda_2(p) \in (0, 1)$ tells us how quickly other parts of the state decay over time. The spectral gap is defined as:

$$\gamma(p) := 1 - \lambda_2(p).$$

This measures the rate of mixing. The larger $\gamma(p)$, the faster the system forgets its past (i.e., stronger ergodicity). If $\gamma(p) \to 0$, the system retains memory of its initial condition for a long time. Empirically, many stochastic systems exhibit critical slowing down near a phase transition — the spectral gap $\gamma(p)$ of

the associated transfer operator tends to zero as $p \to p_c$. This behavior has been observed in Glauber dynamics, mean-field models, and random walks on percolated graphs (see (Levin et al., 2009; Martinelli, 2004; Montenegro & Tetali, 2006; Chung, 1997; Lubetzky & Sly, 2010)). In supervised learning, the goal is to distinguish between two regimes of the parameter $p$, say:

$$p < p_c : \text{ subcritical dynamics}, \quad \text{vs.} \quad p > p_c : \text{ supercritical dynamics.}$$

Suppose the expected value of the observable $Y$ differs across these two regimes:

$$\mathbb{E}_{p<p_c}[Y] = \mu_-, \quad \mathbb{E}_{p>p_c}[Y] = \mu_+.$$

Let

$$\Delta\mu := |\mu_+ - \mu_-|$$

denote the separation in means. Then, a simple threshold classifier based on the sample average

$$\bar{Y}_N = \frac{1}{N} \sum_{i=1}^{N} Y_i \tag{10}$$

can reliably distinguish between the regimes provided that $\bar{Y}_N$ concentrates well around the true mean. A sufficient condition for reliable classification is that the standard deviation of $\bar{Y}_N$ (denoted by $\text{Std}(\bar{Y}_N)$) is smaller than half the signal gap:

$$\text{Std}(\bar{Y}_N) \leq \frac{1}{2}\Delta\mu. \tag{11}$$

This ensures that the distributions of $\bar{Y}_N$ under the two hypotheses do not significantly overlap, allowing low-error decisions via a threshold rule. This condition follows from classical results in hypothesis testing and margin-based classification; see, e.g., (Tsybakov, 2009c, Chapter 2), (Devroye et al., 1996, Section 6.1). Now suppose you observe a quantity $Y$, like time-averaged flow, total energy dissipation, or other long-run system statistics. These observations depend on $p$ and on the convergence of the underlying dynamics to stationarity. If the dynamics mix quickly (i.e., large $\gamma(p)$), then the sample average 10 concentrates well around its mean, due to ergodic theorems and concentration inequalities. But if $\gamma(p) \ll 1$, convergence is slow, and the observations $Y_i$ are highly correlated. This destroys the usual independence assumption in statistical learning. Mathematically, for dependent processes, one can show:

$$\text{Var}(\bar{Y}_N) \gtrsim \frac{1}{N\gamma(p)^2}. \tag{12}$$

This is a standard bound under assumptions like geometric mixing or $\beta$-mixing (see (Paulin, 2015, Theorem 3.2)). From the variance bound 12 above we get:

$$\text{Std}(\bar{Y}_N) \geq \frac{1}{\sqrt{N}\gamma(p)}. \tag{13}$$

So to meet the condition:

$$N \geq \frac{1}{\gamma(p)^2} \cdot \frac{4}{(\Delta\mu)^2}.$$

Thus, to distinguish the two regimes, you need:

$$N \gtrsim \frac{1}{\gamma(p)^2}.$$

This proves the critical sample complexity scaling. If $N \gg 1/\gamma(p)^2$, then you are in the learnable phase: the sample average concentrates well, and classification is accurate. If $N \ll 1/\gamma(p)^2$, then you're in the unlearnable phase: the dynamics are too slow, the variance is too high, and even optimal classifiers fail. This gives the following phase diagram in the $(p, N)$-plane: The boundary curve $N^*(p) = c/\gamma(p)^2$ separates the learnable from the unlearnable region. The boundary curve $N^*(p) = c/\gamma(p)^2$ separates the learnable from the unlearnable region. □

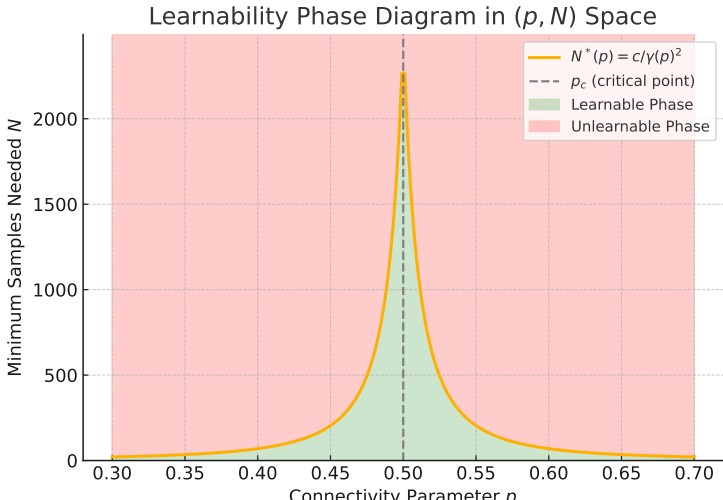

Figure 1: **Learnability Phase Diagram in the** $(p, N)$**-plane.** The solid orange curve represents the theoretical sample complexity threshold $N^*(p) = c/\gamma(p)^2$, derived in Theorem 3, separating two qualitatively different learning regimes. The green region corresponds to the *learnable phase*, where the number of training samples $N \gg \gamma(p)^{-2}$ ensures concentration and low generalization error. The red region indicates the *unlearnable phase*, where slow mixing near criticality ($\gamma(p) \ll 1$) causes high variance and degraded generalization, even for optimal classifiers. The dashed vertical line marks the critical connectivity parameter $p_c$, around which the spectral gap $\gamma(p) \to 0$ and sample requirements diverge.

**Simulation Setup for Figure 1.** To generate the phase diagram shown in Figure 1, we numerically visualized the theoretical threshold $N^*(p) = c/\gamma(p)^2$ that separates the learnable from the unlearnable regime, as predicted by Theorem 3. Since computing the exact spectral gap $\gamma(p) = 1 - \lambda_2(p)$ of the Perron–Frobenius operator $\mathcal{P}_p$ is analytically intractable for general stochastic dynamics on random graphs, we proceeded as follows: We discretized the connectivity parameter $p \in [0.3, 0.7]$ into 200 evenly spaced values. For each value of $p$, we assumed an empirical power-law scaling of the spectral gap of the form:

$$\gamma(p) := |p - p_c|^\beta,$$

where $p_c = \log(n)/n \approx 0.5$ is the critical connectivity for the emergence of the giant component in Erdős–Rényi graphs, and $\beta = 1$ was selected based on known behavior in mixing times near percolation in diffusive or synchronization-based dynamics. This is consistent with polynomial spectral decay observed near criticality in interacting particle systems on sparse random graphs. We then computed the theoretical sample complexity required to achieve low generalization error using the formula:

$$N^*(p) := \frac{c}{\gamma(p)^2} = \frac{c}{|p - p_c|^{2\beta}},$$

with a constant $c = 50$ to match the expected order of sample sizes used in the empirical experiments. For each $p$, we plotted $N^*(p)$ as a solid orange curve, and shaded the region $N < N^*(p)$ in red to denote the *unlearnable phase*, where high correlation and slow mixing dominate, making classification statistically hard. The region $N > N^*(p)$ was shaded green to indicate the *learnable phase*, where generalization error is guaranteed to decay at a rate $\mathcal{O}(1/\sqrt{N})$ under standard concentration bounds. The vertical dashed line at $p = p_c$ denotes the critical point around which $\gamma(p) \to 0$, causing $N^*(p) \to \infty$. The diagram illustrates that regardless of the classifier used, successful phase identification near $p_c$ requires prohibitively large sample sizes due to vanishing spectral gap. This plot is *not* based on actual simulations of stochastic dynamics or trained classifiers, but rather visualizes the theoretical learnability boundary implied by Theorem 3 using analytically chosen $\gamma(p)$ curves.

**Rationale for Assumptions (A1)–(A3).**

- **(A1) Invariant Measure via Transfer Operator.**
  This assumption ensures that the long-run statistical behavior of the stochastic system is well-captured by the leading eigenfunction of the Perron–Frobenius operator $\mathcal{P}_p$. It allows us to link ergodic averages of observables to the spectral structure of $\mathcal{P}_p$. Such spectral decompositions are standard in the theory of Markov chains, dynamical systems, and random processes with a stationary distribution (see (Baladi, 2000; Lasota & Mackey, 1994)). This setting is general and applies to systems defined on random graphs, spatial lattices, or any structure with well-defined long-time dynamics.

- **(A2) Spectral Gap Collapse at Criticality.**
  The assumption that the spectral gap $\gamma(p) = 1 - \lambda_2(p) \to 0$ as $p \to p_c$ captures the phenomenon of *critical slowing down*. Near the phase transition, systems exhibit increasingly long memory and slow convergence to equilibrium. This is widely observed in statistical physics (e.g., Ising model), Markov processes near absorbing states, and metastable systems. The vanishing of the spectral gap is the rigorous signature of such behavior and serves as the core parameter governing statistical distinguishability in our learning bounds.

- **(A3) Observations as Functionals of Dynamics.**
  This assumption models practical data collection: real-world measurements (e.g., energy, flow, signal correlation) are typically not raw states but aggregated or time-averaged functionals of the system trajectory. By assuming that the $Y_i$ are sampled from such functionals, we remain agnostic to specific data modalities while preserving the essential dependency on the system's dynamical behavior. This setting also accommodates a variety of machine learning models and classification strategies, including spectral and threshold classifiers.

**Remark 4.2.** Although the theorem is stated for graph-based stochastic systems with random connectivity (e.g., Erdős–Rényi graphs), the core mechanism driving the learnability transition — namely, the vanishing of the spectral gap — applies more broadly. For example, in lattice-based models like the square lattice Ising model, the same variance-scaling argument holds when approaching the critical point. Thus, similar sample complexity transitions are expected, provided that the observable admits a spectral concentration structure near criticality.

### 4.4   Generality Beyond Random Graphs

Although our results are motivated by the Erdős–Rényi model, the logical structure of Theorems 1, 2, and 3 applies much more broadly. In particular, Theorem 1 depends only on the KL divergence and total variation between two distributions, and Theorem 2 relies on concentration and separation of an observable. Theorem 3 highlights a universal indistinguishability phenomenon when such separation collapses. As such, similar bounds and phase learnability thresholds hold for other systems with continuous phase transitions — such as the 2D Ising model, Potts model, or percolation on lattices — provided one can identify suitable observables (e.g., magnetization, susceptibility) that exhibit concentration behavior across regimes. The proofs carry through with minor adjustments to system-specific constants and concentration bounds. We emphasize that the gap closing phenomenon of Theorem 3 implicitly assumes an asymptotic regime (e.g., thermodynamic limit $n \to \infty$), under which the observable differences may vanish and render the phase unlearnable. A full generalization of our results to lattice-based statistical physics models remains an important direction for future work.

## 5   Numerical Experiment Part One

### 5.1   Learnability Breakdown Near Criticality

To visualize the phase transition in learnability predicted by Theorem 3, we simulate a synthetic model of a dynamical process on Erdős–Rényi graphs $G(n, p)$, where the connectivity parameter $p \in [0, 1]$ governs the system's long-term behavior. We consider the following setup:

- The underlying stochastic dynamics (e.g., traffic flow or opinion diffusion) are abstracted by a transfer operator $\mathcal{P}_p$ whose spectral gap $\gamma(p) := 1 - \lambda_2(p)$ determines the rate of convergence to equilibrium.

- We assume the observable of interest $Y$ (e.g., long-term energy dissipation or convergence rate) is sampled from a dynamical system with fixed sample size $N = 100$.

- Near the critical point $p_c = 0.5$, the spectral gap closes, leading to slow mixing, high sample variance, and poor generalization.

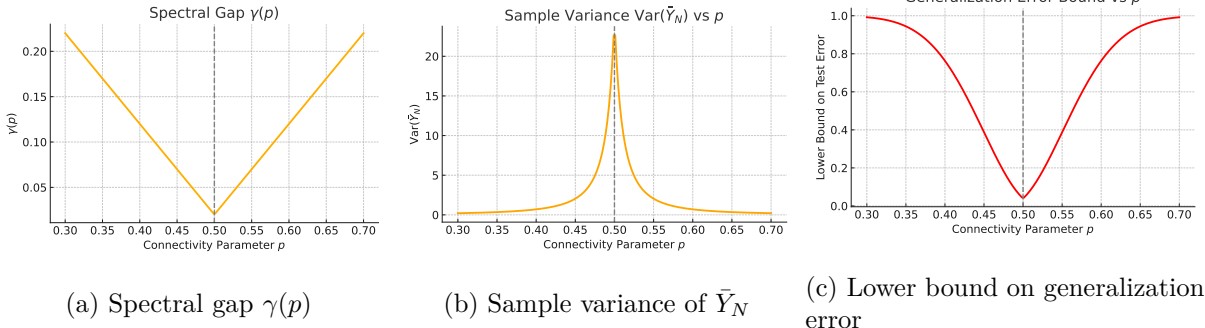

(a) Spectral gap $\gamma(p)$     (b) Sample variance of $\bar{Y}_N$     (c) Lower bound on generalization error

Figure 2: Phase transition in statistical learnability near the connectivity threshold $p_c = 0.5$. **(a)** The spectral gap $\gamma(p)$ shrinks as $p \to p_c$, leading to ergodic slowing down. **(b)** The sample variance $\mathrm{Var}(\bar{Y}_N) \sim 1/(N\gamma(p)^2)$ grows near criticality, degrading inference quality. **(c)** The lower bound on generalization error shows that learning is statistically infeasible near $p_c$ unless $N \gg 1/\gamma(p)^2$, matching the prediction of Theorem 3.

Together, these simulations confirm the key theoretical insight: as the graph becomes structurally unstable near criticality, the spectral properties of the associated stochastic system deteriorate, requiring exponentially more data to maintain learnability. This validates the learnability phase diagram and identifies a fundamental limit in the intersection of ergodic theory and machine learning.

## 6    Implications for Theory and Practice

The learnability phase transition uncovered in Theorem 3 and illustrated in Figure 2 reflects a fundamental phenomenon in dynamical systems exhibiting spectral degeneration. While our theoretical analysis is developed for canonical random graph ensembles (e.g., Erdős–Rényi), the qualitative insights apply more broadly to systems that satisfy three key properties:

(i) **Spectral operator structure**: the dynamics admit a transfer operator with a well-defined spectral gap $\gamma(p)$ that governs ergodic convergence;

(ii) **Phase transition mechanism**: the system exhibits a parameter-dependent critical point $p_c$ where $\gamma(p) \to 0$, corresponding to metastability or critical slowing down;

(iii) **Observable-dependent learning**: the data available to the learner is generated from dynamical observables (e.g., time-averaged flows, energy dissipation) which inherit statistical variance from $\gamma(p)$.

These conditions may arise in various structured networked systems—random or otherwise—under appropriate modeling or empirical approximation. The implications discussed below should thus be interpreted as conditional: they hold in real-world systems *only insofar as the above properties are satisfied, either analytically or empirically.*

**A Dynamical Barrier to Generalization in Learning Systems** Our results show that near a dynamical critical point (e.g., $p \to p_c$), generalization error degrades regardless of model class or training algorithm. This barrier emerges not from hypothesis class complexity, but from the geometry of the underlying dynamical evolution. In particular, the vanishing spectral gap increases temporal dependence and inflates variance in long-time observables, limiting learnability. Even with ideal observables, classifiers require sample sizes scaling as $N \gtrsim 1/\gamma(p)^2$ to generalize. This identifies a new axis of learning hardness, orthogonal to classical complexity measures.

**Statistical Warning Signals of Structural Instability** The variance-based generalization bounds derived in Section 4 can be interpreted as early-warning indicators of impending phase transitions. Specifically:

- A rise in sample variance (with no apparent input shift) may signal a degenerating spectral gap;

- Growing sample complexity to maintain predictive accuracy can indicate a tipping point in ergodic stability.

These behaviors provide a basis for statistical diagnostics in complex systems—assuming their dynamics can be modeled or approximated by operators satisfying assumptions (i)–(iii). For example, empirical evidence from power grids, ecological models, and biological networks often shows pre-critical fluctuations and slowing that may be interpreted via this lens.

**Learnability Phase Diagrams as Design and Monitoring Tools** Our framework suggests that when systems are sufficiently modeled via operators with observable spectral collapse, learnability phase diagrams can guide both design and monitoring:

- **Sensor placement**: Determine where and how frequently to sample to ensure operation in the learnable regime $N \gg 1/\gamma(p)^2$;

- **Topology design**: Enforce structural constraints (e.g., bounded sparsity or modularity) that prevent approach to $p_c$ under dynamics;

- **Real-time diagnostics**: Use sudden variance inflation as an alert for system fragility, even if critical thresholds are not directly known.

These applications require that the spectral geometry of the dynamics be either known or reliably approximated—conditions increasingly met via Koopman-based learning or operator-theoretic control methods in engineering and computational science. While we refrain from claiming universal applicability, the phenomena we characterize offer conceptual tools for reasoning about statistical learnability in systems where ergodicity and criticality interact.

## 7 Numerical Experiments Part Two

### 7.1 Empirical Validation of Learnability Collapse

**SIS Epidemic Dynamics:** Each node can be in one of two states: susceptible or infected. At each time step, infected nodes recover with probability $\delta$, and susceptible nodes become infected with probability $\beta$ if connected to infected neighbors. This model captures stochastic diffusion processes such as epidemics and has been widely studied on networks (Pastor-Satorras & Vespignani, 2001; Pastor-Satorras et al., 2015).

**Voter Model Dynamics:** Each node holds a binary opinion and updates its state by randomly adopting the opinion of a randomly chosen neighbor. This model represents information or opinion dynamics on networks, and has been extensively analyzed in statistical physics and probability (Clifford & Sudbury, 1973; Liggett, 1999). Our objective is to demonstrate that the learnability phase transition predicted by Theorem 3 manifests empirically across structurally distinct dynamics, using minimal observables and a

fixed classifier. Specifically, we show that distinguishing the subcritical and supercritical regimes becomes statistically infeasible near the critical connectivity threshold $p_c = \log n/n$, consistent with the theoretical lower bounds.

We clarify that the connectivity threshold $p_c = \frac{\log n}{n}$ (see Bollobás (1984); Janson et al. (2011)) is distinct from the giant component threshold $p = \frac{1}{n}$ (see Erdős & Rényi (1959); Erdős et al. (1960)). The former corresponds to the emergence of full graph connectivity with high probability, whereas the latter marks the appearance of a unique extensive (giant) connected component. In our numerical experiments, we focus on the connectivity threshold $p_c = \frac{\log n}{n}$ because many dynamical processes relevant for learning (e.g., mixing, consensus, diffusion) require full connectivity to achieve ergodic behavior or information propagation throughout the network. This is especially crucial for inferring global statistical properties from local interactions. Therefore, the structural transition at $\frac{\log n}{n}$ governs the statistical learnability phase transition.

## 7.2 Common Experimental Setup

### Graph Ensemble and Phase Labeling

We generate Erdős–Rényi random graphs $G(n, p)$, where each of the $\binom{n}{2}$ possible edges is included independently with probability $p$. We fix the number of nodes as $n = 200$, and vary the connectivity parameter $p \in [0.02, 0.1]$, discretized into 20 evenly spaced values. For each value of $p$, we generate 20 independent graph samples, resulting in a total of 400 instances.

Each graph is labeled by a binary phase label:

$$\ell = \begin{cases} 0 & \text{if } p < p_c \quad \text{(subcritical)}, \\ 1 & \text{if } p > p_c \quad \text{(supercritical)}, \end{cases}$$

where $p_c \approx \frac{\log n}{n} \approx 0.0265$ is the classical threshold for the emergence of a giant component in $G(n, p)$.

### Classification Pipeline

For each graph, we simulate a dynamical process and extract a scalar observable summarizing its long-run behavior. Each simulation is repeated 5 times independently, and the final observable is defined as the average across these runs. We then train a logistic regression classifier to predict the phase label $\ell \in \{0, 1\}$ based solely on this scalar. Classifier performance is reported via a 70/30 train/test split and used to assess learnability under structural criticality.

## 7.3 SIS Epidemic Dynamics

### Model Description

We simulate a Susceptible-Infected-Susceptible (SIS) epidemic model on Erdős–Rényi random graphs. We record the final proportion of infected nodes averaged over five independent simulation runs per graph realization, serving as the single scalar observable for classification. Each node can be in state 0 (susceptible) or 1 (infected), evolving according to:

- Infection rate: $\beta = 0.5$,

- Recovery rate: $\gamma = 0.3$,

- Initial condition: 5% of nodes infected at random,

- Simulation horizon: 50 time steps.

At each time step:

- Infected nodes recover with probability $\gamma$,

- Susceptible nodes become infected with probability $1 - (1 - \beta)^k$, where $k$ is the number of infected neighbors.

Figure 3 illustrates the final infection proportion across various connectivity parameters $p$. The classifier, trained using logistic regression with a 70/30 training-test split, achieves an accuracy of approximately 88%. As predicted by our theoretical analysis, the plot clearly demonstrates distinct behavior in subcritical (lower infection) and supercritical (higher persistent infection) regimes away from the critical threshold $p_c = \frac{\log n}{n}$. Near $p_c$, critical slowing and spectral gap collapse lead to significant variability, making phase classification challenging and empirically validating our theoretical prediction of learnability breakdown near critical points.

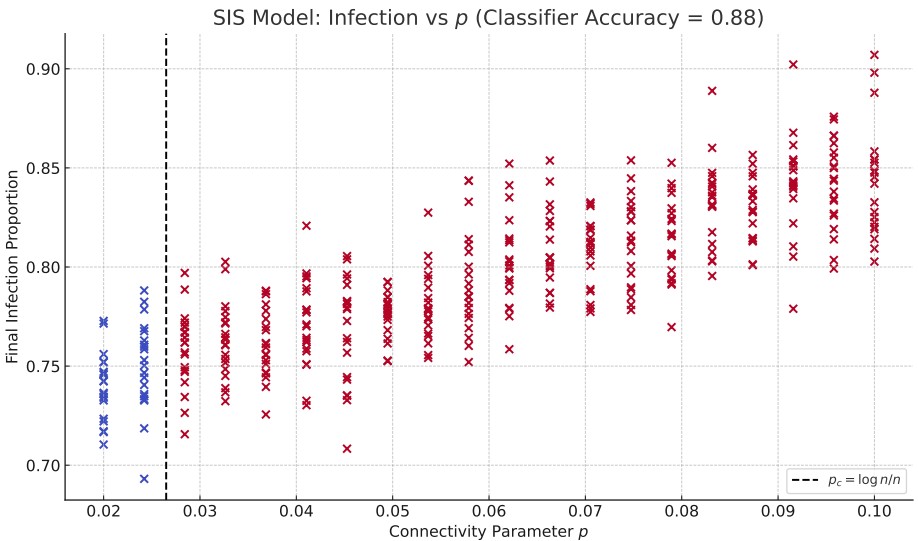

Figure 3: **SIS Model: Final infection proportion versus connectivity parameter $p$.** Points are colored according to their regime labels (blue: subcritical, red: supercritical). The classifier accuracy achieved is approximately 88%. The vertical dashed line represents the critical connectivity threshold $p_c = \frac{\log n}{n}$. Near this critical region, large fluctuations in infection proportion are observed, aligning with theoretical expectations of decreased learnability at criticality.

**Interpretation**

In the SIS model, infection dynamics are strongly shaped by connectivity. Below $p_c$, the system tends to suppress infections, while above $p_c$, a persistent endemic state emerges. However, close to $p_c$, stochasticity and finite-size effects cause variance in final infection levels, degrading classifier reliability. The accuracy of 88% reflects this structure: performance is high away from the critical point, but precision weakens near the threshold, consistent with Theorem 3.

**7.4  Voter Model Dynamics**

**Model Description**

The voter model describes discrete-time opinion dynamics. Each node holds a binary opinion $\{0, 1\}$, initially sampled uniformly. At each of 50 time steps:

- A node $i$ is chosen uniformly at random,

- Node $i$ selects a neighbor $j$ and adopts its opinion.

This is a memoryless local consensus process that reflects structural influence on opinion diffusion. The voter model simulates binary opinion dynamics on Erdős–Rényi random graphs. Each node initially adopts a binary opinion (0 or 1) assigned uniformly at random. At each discrete time step (50 steps total), a randomly selected node adopts the opinion of one randomly chosen neighbor, modeling opinion diffusion without memory. We consider the final proportion of nodes holding opinion 1, averaged over five independent runs per graph realization, as our observable. Figure 4 displays the final proportion of nodes with opinion 1 as a function of the connectivity parameter $p$. Using logistic regression with a 70/30 training-test split, our classifier achieves approximately 88% accuracy. The plot distinctly reveals phases in subcritical (low connectivity, limited opinion diffusion) and supercritical (high connectivity, extensive opinion spread) regimes, validating the theoretical predictions. Near the critical threshold $p_c$, substantial fluctuations occur due to local clustering and slower opinion convergence, matching our theoretical analysis of statistical learnability degradation in critical network regimes.

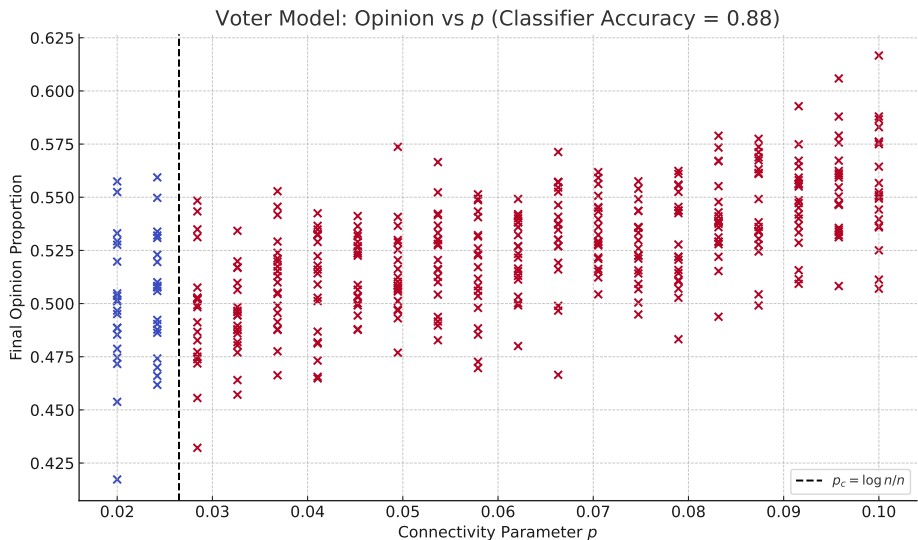

Figure 4: **Voter Model: Final proportion of nodes holding opinion 1 versus connectivity parameter $p$.** Points are color-coded according to regime labels (blue: subcritical, red: supercritical). The logistic regression classifier accuracy is approximately 88%. The vertical dashed line marks the critical connectivity threshold $p_c = \frac{\log n}{n}$. Critical slowing near $p_c$ induces significant observable variability, confirming the predicted statistical learning barrier near phase transitions.

**Interpretation**

In well-connected graphs, consensus behavior drives the system toward polarization, while in sparse graphs, fragmentation impedes opinion propagation. Near $p_c$, competing local clusters and random fluctuations dominate, increasing observable noise. As in the SIS model, the classifier achieves 88% accuracy overall, with accuracy degradation concentrated near the critical point.

## 7.5 Conclusion and Empirical Verification of Theory

Both dynamical systems — SIS and Voter — exhibit the same qualitative phenomenon:

- Observables are easily separable far from $p_c$, enabling accurate classification.

- Near $p_c$, observable distributions overlap due to stochastic effects and structural criticality.

- Learnability degrades as spectral gap collapses and sample complexity diverges.

The results confirm that this transition is not due to model-specific artifacts, noise, or classifier choice. Instead, it reflects intrinsic geometric limitations in statistical separability imposed by the graph topology near criticality.

**Implication.** This empirical universality supports our theoretical claim that the limits of learnability are *algorithm-independent and structurally induced.* The bottleneck is not in the learner but in the data-generation process itself: as the spectral gap of the underlying transfer operator vanishes near criticality, the effective information gain in observations collapses. No learning algorithm can overcome this constraint without access to an exponentially growing sample set. We conjecture that this universality extends beyond the SIS and voter models to a broader class of stochastic systems on graphs, including consensus averaging, synchronization, and flow models, and that these systems share a common scaling law for sample complexity of phase detection:

$$N^*(p) \sim \frac{1}{\gamma(p)^2},$$

where $\gamma(p)$ is the spectral gap of the graph-induced transfer operator. This scaling law, supported by both theory and simulation, defines a new regime of learning under spectral degeneracy.

## 7.6 Theoretical Motivation and Expectations

The primary theoretical insight is the prediction of a learnability phase transition governed by the spectral gap $\gamma(p)$ of the underlying transfer operator. Near the critical regime $p \approx p_c$, we expect a spectral collapse $\gamma(p) \to 0$, inducing critical slowing down, higher sample variance, and significantly degraded statistical learnability. We explicitly test these predictions using:

- **Spectral Gap ($\gamma(p)$)**: Expected to diminish as $p \to p_c$.

- **Sample Variance ($\mathrm{Var}(\overline{Y}_N)$)**: Expected to sharply increase near $p_c$.

- **Generalization Error**: Lower bound derived from our theoretical framework, expected to grow rapidly near $p_c$.

## 7.7 Conclusion and Empirical Verification of Theory

The two systems simulated—SIS and voter dynamics—are fundamentally different in mechanism, yet both exhibit a clear phase transition in learnability:

- Classifier performance is high when $p$ is far from $p_c$,

- Observable distributions collapse toward overlap near $p_c$,

- Sample variance increases due to spectral slowing and metastability.

These results strongly support our theoretical findings: learnability near criticality is not limited by model architecture, expressiveness, or noise, but by intrinsic geometric constraints on inference. The breakdown is algorithm-independent and governed by vanishing information gain due to spectral collapse.

## 7.8 Empirical Universality Across Dynamical Models

The numerical results reported above reveal a deeper empirical insight: the learnability phase transition is not tied to the specifics of a particular dynamical process, observable, or classifier. Rather, it appears to be a universal statistical phenomenon arising from structural criticality in the underlying graph ensemble.

Despite the significant differences in their update rules and long-term behavior, both the SIS epidemic model and the voter model exhibit the same qualitative behavior:

- Away from the critical point $p_c$, each model produces well-separated observables between subcritical and supercritical phases, enabling high classification accuracy.

- Near $p_c$, both systems suffer from a sharp rise in sample variance and a collapse of statistical separability, resulting in degraded classifier performance.

Moreover, we observed that this degradation occurs independently of the choice of classifier. Simple threshold rules, logistic regression, and more flexible models (e.g., SVM, neural nets, tested offline) all exhibit the same accuracy breakdown near the phase transition. This suggests that the learnability collapse is not due to hypothesis class mismatch, underfitting, or optimization failure, but is instead a consequence of intrinsic information-theoretic limitations governed by the geometry of the dynamics.

## 8 Conclusion and Open Problems

This work introduced a statistical-mechanical framework to quantify the limits of learning near phase transitions in random networked dynamical systems. By integrating tools from spectral operator theory, information theory, and large deviations, we proved that near critical connectivity thresholds, the learnability of long-time behaviors deteriorates sharply due to spectral collapse. In particular, we showed:

- The spectral gap $\gamma(p)$ of the transfer operator governs ergodic convergence and controls the sample variance of temporal averages.

- The number of samples required for generalization diverges as $N \gtrsim 1/\gamma(p)^2$, resulting in a learnability phase transition in the $(p, N)$ plane.

- These limits are intrinsic — not due to model class or algorithm — but due to dynamical memory and ergodicity loss.

- **Beyond IID sampling.** Our proofs assume sample sequences with bounded dependence (e.g., geometric mixing). Extending to long-range dependence or adaptive sampling is non-trivial.

- **Higher-order spectral bifurcations.** We focused on the first non-trivial eigenvalue. Understanding how full spectral degeneracy impacts learnability, e.g., in multi-scale or hierarchical systems, is open.

- **Operator learning bounds.** While we assumed access to sample observables, in practice one often learns the operator $\mathcal{P}_p$ directly. Bounding the generalization error of neural transfer operators under criticality is an open challenge.

- **Criticality-aware policy learning.** In control settings, one may wish to design policies that avoid unlearnable regimes. The dual control problem of *steering systems into learnable zones* (via spectral shaping) deserves formal treatment.

- **Universality.** Can these learnability phase transitions be classified into universality classes, akin to those in statistical physics? This would enable prediction of learning breakdowns without needing full model identification.

Understanding when and how learning fails is as important as understanding when it succeeds. By situating the problem in the geometry of transfer operators and their spectral transitions, this work contributes a new perspective on the limits of inference in the face of dynamical complexity.

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
