# OpenReview forum: "Learning Criticality: Statistical Limits of Predicting Phase Transitions in Random Networks"
_TMLR — Rejected by TMLR_

### Review · Reviewer_o41S · 2025-06-23

**Summary Of Contributions:**

The manuscript systematically explores the statistical limits of learning phase transitions from samples of systems in random networks. It aims to determine, from probabilistic and information-theoretic perspectives, the feasibility of inferring a system's critical state from finite samples. Specifically, it examines whether one can accurately assess if a system is subcritical or supercritical based on observational data from stochastic processes on random graphs near a critical threshold.  The authors give a general information-theoretic lower bound on sample complexity for distinguishing subcritical and supercritical regimes, applicable to any learning algorithm.
Specifically, by integrating tools from several fields, they proved that near critical connectivity thresholds, the learnability of long-time behaviors deteriorates sharply due to spectral collapse, the spectral gap of the transfer operator governs ergodic convergence and controls the sample variance of temporal averages. The number of samples required for generalization diverges, and it leads to an intrinsic learnability phase transition.

**Audience:**

No

**Broader Impact Concerns:**

I believe the research would be of interest to a broader audience, including those in the fields of statistical inference, complex networks, and beyond.

**Claims And Evidence:**

Yes

**Requested Changes:**

I request to add more numerical simulation results on different systems and perhaps using different learning algorithms to support the theoretical analysis.

**Strengths And Weaknesses:**

Strengths: The obtained result is an intrinsic learnability phase transition, regardless of the model and the inference algorithm.
Weakness: The numerical results are not significant enough.

---

### Review · Reviewer_N9NW · 2025-07-11

**Summary Of Contributions:**

The paper aims at providing a formal framework for statistically learning phase transitions in
random networked systems. Concretely, the problem is formulated as a binary classification
problem, where the learner has to infer whether an unknown graph parameter $p$ lies below or
above a threshold $p_c$. The learner is given a set of $N$ i.i.d. samples of observable features
associated with the graph. The setting of interest involves data drawn from two unknown
probability distributions on the observable features, which are defined within a window on both
sides of the threshold. The goal is to study how the sample complexity behaves around the critical
region as well as away from it.

The main results of the work are summarized in three theorems: the first theorem establishes a
minimax lower bound and is a known result in the broader context of binary hypothesis testing. This result is used here
to show how the sample complexity diverges as the Kullback-Leibler divergence between the
aforementioned distributions vanishes. The second theorem concerns the case in which a scalar
observable used to describe the system sharply concentrates on two well-separated values on
each side of the threshold (akin to an order parameter). This theorem, which also builds on
known results in statistical learning, puts an upper bound on the generalization error and allows to understand that
the sample complexity — using a hard-threshold classifier — becomes more amenable as such
concentration occurs. Finally, the third result uses the framework of spectral theory in dynamical
systems to derive, from a different angle, how the sample complexity diverges as the critical point
is approach, inversely proportional to the square of the gap of a transfer operator. The theoretical
results above are complemented by numerical simulations of various dynamics on random
graphs.

Overall, this work provides a framework to formalize physical intuition and empirical observations
about statistically learning continuous phase transitions, building on classical results in learning theory, statistical inference, and the transfer operator formalism.

**Audience:**

Yes

**Broader Impact Concerns:**

There are no broader impact concerns.

**Claims And Evidence:**

No

**Requested Changes:**

**The questions and changes that I consider critical for making my recommendation are the following:**

• While I like the reasoning in the proof of Theorem 3 to derive the sample complexity scaling as
in terms of the spectral gap, in my opinion, further details should be provided to understand the bounds on the
generalization error that are mentioned in the statement of the theorem. Furthermore, the
authors should be more careful stating what the different parameters mean in this theorem. For
example, what is $\delta$ in the generalization error bound in the bullet-list point corresponding
to $|p - p_c| > \Delta$?

• Overall, the three theorems in Sec. 4, seem to be rather generic, not depending crucially on
having considered random graphs. Am I right to assume that the same results would apply —
with minimal modifications in the hypotheses — for example, when trying to learn the phases of
the square lattice Ising model (or, for that matter, any other system featuring a continuous phase transition)? I
would appreciate if this is better discussed in the paper.

• My main concern is about Sec. 7 and the interpretation of the numerical results presented there. For example,
for the Epidemic dynamics model, it is stated that “below $p_c$, the system quickly
extinguishes infections”, but actually the final infection proportion shown in Fig. 5, is still above
68% in the subcritical region. So from the numerics one cannot really appreciate what it’s
stated in the interpretation of such results. Likewise for the Voter model, where in Fig.
6 one can see that the final opinion proportion fluctuates around 50% both below and above
the critical threshold. Further, the size of the fluctuations is roughly the same throughout the
whole range of $p$ values, as opposed to the claim that the “Critical regime exhibits increased
observable variance”. In my opinion, the conclusions presented in Secs. 7.4 and 7.5 are not
fully supported by the numerical results reported in Sec. 7. The fact that the numerical results
do not entirely match the theoretical expectations might be due to a number of different
factors: finite-size effects, having constrained the range of $p$ values only to the critical region (for these examples, what is the critical window $\Delta$ of Theorem 3 that separates the learnability phases?), or not having reached a true steady-state solution— all
of these factors could be systematically and easily checked by carrying out more simulations,
which is what I would strongly advice the authors to do. I would also strongly advice the
authors to revise their interpretation and conclusions in a way that better reflects their empirical
results.

**The following comments and questions should help strengthen the presentation and content of
the work:**

• I would recommend to elaborate more on Fano’s inequality, Vapnik’s
inequality and Rademacher complexity bounds, for example in Sec. 3, since these results are the basis for the proofs of Theorems 1 and 2.

• In Sec. 3, the discussion of the phase transition in the Erdős–Rényi model is treated a bit too
lightly. In particular, the authors should mentioned that the discussed features are asymptotic
statistical properties for random graphs in the limit $n \to \infty$, which is needed to rigorously
speak of phase transitions. In this respect, are the main results in Sec. 4 modified when
formally taking the thermodynamic limit? Is it correct to say that the gap closing hypothesis in
Theorem 3 implicitly assumes a thermodynamic limit?

• It not clear what results Sec. 5 refers to. Is it to the
plots in Fig. 4? If so, this should be explicitly mentioned in the text.

• Regarding Fig. 4, in my opinion, the authors should include more details on the simulations
carried out there. What is the system size considered here? Is the plot of the variance in
subfigure (b) obtained from their numerics, or is it the plot of the theoretical result used in
Theorem 3? If it’s the former, they could add the plot of the latter for comparison. Adding curves for at least one more value of the sample size $N$ would also be desirable.

• The labelling of the plots should be amended: subfigures (a), (b), and (c) in Fig. 4 are labelled as
Figs. 1, 2, and 3, respectively.

• In the numerical simulations discussed in Sec. 7, the authors refer to the critical connectivity
threshold of the Erdős–Rényi model, $p_c = \log(n)/n$. However, before this point, all
considerations were made for the critical threshold for having a large (extensive) connected
component, which is, $p_c = 1/n$. Can the authors clarify why they have considered in this
section a different threshold? Along these lines, the authors should correct the sentence in Sec.
7.1, which reads: “where $p_c \approx \log n /n \approx 0.0265$ is the classical connectivity
threshold for emergence of a giant component in $G(n,p)$”, as the thresholds for
connectedness and for having a giant component, are two distinct thresholds.

• In Figs. 5 and 6, the authors could plot on top of all the markers shown, the observable sample
mean for each value of the parameter $p$, in order to better appreciate the trend of the mean
as well as the variability (the latter been already shown).

• The test-set accuracy reported in the text for the SIS Epidemic Dynamics analysis is 92%.
However, in the plot in Fig. 5, it is written “classifier accuracy = 0.88”. Can the authors clarify
which of those is the correct value? Same consideration applies for Fig. 6 for the Voter model
dynamics, where it also reads “classifier accuracy = 0.88”.

• To support the statement at the end of Sec 7.4, that “These results strongly support our
theoretical findings: learnability near criticality is not limited by model architecture,
expressiveness, …”, I think it would be nice to actually support that statement by showing
consistent numerical results with at least one more model, since as far as I can tell, they have
only considered a logistic regression classifier.

• Proper citation is also needed with respect to:
- Phase transitions in the Erdös-Rényi random graph model (at the beginning of Sec. 3).
- When introducing the models for the numerical simulations in Sec. 7.

**Strengths And Weaknesses:**

**Strengths**

The paper is well written and organized. I also think that the analysis to derive the critical scaling of the sample complexity in terms of the spectral gap, provided in Theorem 3, is very instructive. Furthermore, I believe that the potential future research lines pointed out in Sec. 8, are timely and interesting.

**Weaknesses**

However, I have also identified some weaker aspects, some of which I also address in the Requested Changes part below. In particular, I find the proof of Theorem 3 incomplete and I think more details should be provided to prove the generalization error bounds stated in this theorem.

In Sec. 6, the authors discuss possible implications for real-world systems such as power grids, biological networks, etc. However, it is well know that many of such real-world networks are, in fact, not well described by the random graph model. Thus, strictly speaking it is not clear whether the claims made in that section would indeed apply to the scenarios described there.

The interpretation of the results and conclusions in Sec. 7, in my opinion, do not fully reflect the observations of the numerical examples reported there. Further numerical simulations seem to be needed in order to make a more systematic analysis of these results and validate the authors’ interpretations and theoretical expectations.

---

### Decision · Action_Editor_gJHf · 2025-08-25

**Recommendation:** Reject

**Audience:**

No

**Audience Explanation:**

The scope of the paper is very much oriented towards physics. The author may therefore consider re-submitting in a more specialised physics journal.

**Claims And Evidence:**

No

**Claims Explanation:**

After the reviews were provided, the paper has been almost entirely revised. It is therefore necessary to get new independent reviews to get a fair assessment.